# Preparation and Process Parameter Optimization of Continuous Carbon Fiber-Reinforced Polycarbonate Prepreg Filament

**DOI:** 10.3390/polym15030607

**Published:** 2023-01-24

**Authors:** Xun Chen, Yesong Wang, Manxian Liu, Sheng Qu, Qing Zhang, Shuguang Chen

**Affiliations:** 1College of Automation, Jiangsu University of Science and Technology, Zhenjiang 212000, China; 2School of Mechanical Engineering, Jiangsu University of Science and Technology, Zhenjiang 212000, China; 3School of Aerospace Engineering and Applied Mechanics, Tongji University, Shanghai 200092, China; 4School of Mechanical Engineering, University of Science and Technology Beijing, Beijing 100083, China

**Keywords:** continuous carbon fiber reinforced polycarbonate prepreg filament, continuous fiber-reinforced composite, 3D printing, process parameter optimization

## Abstract

Continuous fiber-reinforced composite 3D printing (CFRC 3DP) has become a hot topic of interest for many experts and scholars. Continuous fiber-reinforced prepreg filament (CFRPF) for printing needs to be prepared in advance. In this paper, on the basis of the resin fusion impregnation theory, a fabrication device was designed for continuous carbon fiber-reinforced polycarbonate prepreg filament (CCFRPF). Then, according to the orthogonal test and the TOPSIS entropy weight optimization theory, the optimization method for CFRPF/PC preparation process parameters was proposed, and the relationship between the preparation process parameters and the performance indexes was discussed. The results show that when preparing CCFRPF/PC, the weight of diameter performance index is the largest, about 0.75. The optimal combination of process parameters for CCFRPF/PC is, respectively, 285 °C for the outlet mold temperature, 305 °C for the impregnation mold temperature, and 1 m/min for the winding speed. In this case, the diameter, roundness, minimum curvature radius and tensile strength of 0.375 mm, 29.4 μm, 9.775 mm and 1298 MPa were achieved, respectively.

## 1. Introduction

Three-dimensional printing is an additive manufacturing (AM) method [1,2] that can produce products rapidly, compared to traditional subtractive manufacturing, is another technological revolution in the manufacturing field. It shows the vitality and potential of personalized creation in the new era, and has been widely used in mold manufacturing, aerospace, automobiles, home appliances, medical and other fields. Among the many 3D printing technologies, fused deposition modelling (FDM) has attracted attention for its low cost and convenient operation [3]. However, due to the limited material capacity, the strength of the printed parts by FDM is low and cannot support excessive loads, which generally cannot meet actual use requirements [4]. With further study of composites, it was found that continuous fiber-reinforced composites (CFRC) had the advantages of better physical [5] and mechanical properties, as well as being lightweight and having a good overall design [6].

Currently, there are two main methods to print continuous fiber-reinforced composites using FDM [7]. Namiki and Tian et al. [8,9] proposed mixing raw silk from continuous fiber and thermoplastic material in the nozzle cavity, and then extruding through the nozzle to print CFRC. This method of FDM printing is called “co-extrusion” CFRC 3DP, as shown in Figure 1a, but continuous raw fiber has some disadvantages such as incomplete impregnation, large porosity of the printed sample, and the lay density of reinforced fiber and thermoplastic matrix materials cannot be controlled according to actual needs. The printing principle of “independent extrusion” CFRC 3DP is shown in Figure 1b. A typical company representative is MarkForged company [10,11], which needs to prepare a continuous fiber-reinforced prepreg filament in advance, and then use two independent printing heads to print thermoplastic materials and CFPF, respectively. This method overcomes the various defects of “co-extrusion” and MarkForged company reports show that the mechanical properties of printed samples are significantly improved.

Different principles of CFRC 3DP use different materials, the continuous fiber raw silk can use “co-extrusion” molding technology directly, with no need to be prepared into prepreg filament in advance. For the “independent extrusion” molding technology studied in this paper, continuous fiber raw silk is presoaked and coated with resin material into CFRPC. Obviously, the preparation of CFRCF requires the use of thermoplastic resin materials and continuous fiber reinforced materials, and to ensure that the thermoplastic resin material can be securely wrapped on the surface of the continuous fiber [12], the resin material should be compatible with the continuous fiber in physical, chemical and thermal properties. Currently, the MarkForged company is one of the most famous companies producing continuous fiber reinforced prepregs filament in the world for CFRC 3DP. The polyamide (PA) resin is suitable for steeping continuous carbon fiber, glass fiber or aramid fiber and the like to be prepared into CFRPF/PA by the MarkForged company [13], and the specific parameters are shown in Table 1. It is known that the Anisoprint Company can provide continuous carbon fiber-reinforced composite materials (CCFRC) and continuous basalt fiber composite materials (CBFRC), and the continuous fibers are obtained by pre-impregnating and solidifying in thermosetting resin materials [14], but detailed parameters have not been shown.

Kuba et al. [11] sent the continuous fiber and granular peek into the heating cavity of the heater, and then extruded the continuous fiber-reinforced peek prepreg filament from the nozzle. Additionally, the different CFRPFs were prepared by controlling the extrusion speed, and performance indexes such as void and strength were evaluated. The results showed that porosity is reduced by 92% compared to standard polymer prepregs, and the tensile strength of the printed sample prepared by CFRPF/PEEK was increased by 116.8%. Kaczmarek et al. [15] prepared CCFRFP/PA from dry 3 K carbon fiber by melting nylon resin, and improvements are suggested in the melting impregnation process based on experimental results and impregnation models. Li et al. [16] prepared CFRPF/PLA with polylactic acid and continuous carbon fiber by melting impregnation, and the diameter of the die outlet, screw speed and traction speed were, respectively, 0.6 mm, 11 r/min and 450 mm/min, and the mechanical performance of the printed PLA composites reinforced with continuous carbon fiber was investigated. Altug et al. [17] designed a polymer impregnation line to produce prepreg forms of continuous fiber-reinforced thermoplastic filaments with different fiber fractions. The PLA filament and continuous carbon fiber were formed together through the mold, and the fiber volume content of CFRPF was controlled by adjustment of the diameter of the outlet, and the mechanical properties and failure modes of the prepared CFRPF prints were evaluated. Cui et al. [18] designed an integrated device for continuous fiber-reinforced thermoplastic 3D printing filament, and prepared continuous glass fiber-reinforced PLA prepreg filament. The prepreg filament was used as 3D printing consumables in the 3D printing equipment of FDM, and the influence of the printing process parameters on the mechanical properties of the sample was evaluated. Ming and Hao et al. [19,20] studied a 3D printing technology of continuous carbon fiber-reinforced thermosetting epoxy resin composites. The continuous fiber was impregnated with molten epoxy prepolymer inside the print head, and the polymer crosslinking reaction was completed through thermal post-curing after printing. CFRPF/PLA was used in 3D printing and is prepared by molten impregnation process, and the orthogonal experiment method was used to study the process parameters of CFRPF/PLA [21]. James et al. [22] designed a prepreg filament and narrow tape for additive manufacturing, but prepreg molds are not efficient in producing CFRPF, and the aging of resin material is serious. To sum up, most scholars evaluate the mechanical properties of the printed parts after the preparation of a specific CFRPF or prepare CFRPF by building a simple device. However, the optimization of the preparation process of a continuous fiber-reinforced prepreg filament is rarely mentioned. If CFRC 3DP technology needs to be developed, optimization of the preparation process of different types of CFRPF will be a key technical issue.

In the last few years, domestic and foreign scholars began to focus on the study of CFRPF preparation technology; due to the lack of unified standards, minority CFRPF can be applied in practice, which becomes the technical bottleneck restricting the further development of “independent extrusion” CFRC 3DP. On the basis of the preparation technology of molten impregnation CFRPF, it is urgent to study the interaction between the process parameters and the performance index of CFRPF, so as to provide a theoretical basis for the preparation standard of CFRPF.

Polycarbonate (PC) is a widely used engineering plastic with high strength, high toughness, excellent electrical insulation and flame retardancy. Additionally, because it is easy to process and color, it is favored by enthusiasts in the field of 3D printing. However, the poor mechanical properties of 3D-printed samples cannot be directly used for industrial applications. In order to meet the requirements of PC performance in more scenarios, 3D printing of continuous carbon fiber-reinforced PC filament (CFRPC/PC) will become a trend. In this paper, CFRPF was prepared by melting impregnation process, with continuous carbon fiber as the reinforcement material and polycarbonate as the impregnation material; the CCFRPF/PC was prepared as an example to propose an optimization method of preparation process parameters based on entropy weight TOPSIS method. The results show that in the preparation of CCFRPF/PC, the diameter performance index weight of CFRPF is the largest, about 0.75. When the temperature of the outlet mold is 285 °C, the temperature of the impregnation mold is 305 °C, and the winding speed is 1 m/min, the process parameter of CCFRPF/PC is optimal. Additionally, the diameter, roundness, minimum curvature radius and tensile strength of 0.375 mm, 29.4 μm, 9.775 mm and 1298 MPa were achieved in the performance indexes of CCFRPF/PC, respectively. In addition, the process optimization strategy studied in this paper can provide a theoretical basis for the preparation of more types of CFRPF.

## 2. Experimental Foundation

The TOPSIS method is a ranking method based on the proximity between a finite number of evaluation objects and an idealized target, which is of great significance for multi-objective decision analysis [23]. Using the TOPSIS decision model, $X=\left\{X_{1},X_{2},X_{3},\cdots ,X_{m}\right\}$, is the set of m alternative schemes, $Y_{i}=\left\{Y_{i1},Y_{i2},Y_{i3},\cdots ,Y_{in}\right\}$ is the index set for the ith alternative scheme, and $Y_{ij}$ is the jth index of the ith scheme.

The index as the objective function, the $Y_{ij}=f\left(X_{i}\right)$, among them, i=1,2,3,⋯, m, j=1,2,3,⋯, m. Then, the sample decision matrix is shown in Equation (1). Equation (2) is used to normalize the decision matrix and form a normalized matrix Z, as shown in Equation (3), $X_{ij}=w_{j}Z_{ij}$, $w_{j}$ are weighting factors.
(1)$$Y={\left(Y_{ij}\right)}_{m\times n}$$
(2)$$Z_{ij}=\frac{Y_{ij}}{\sqrt{\sum_{i=1}^{m}Y_{ij}^{2}}}$$
(3)$$Z=\left[\begin{array}{ccc} X_{11} & \cdots & X_{1m}\\ \vdots & \ddots & \vdots \\ X_{n1} & \cdots & X_{nm} \end{array}\right]$$

In m different design schemes, the positive and negative ideal solutions of the evaluation index are shown in Equations (4) and (5), respectively.
(4)$$Z^{+}=\left(Z_{1}^{+},Z_{2}^{+},\cdots ,Z_{m}^{+}\right)=(\max\left\{X_{11},X_{21},\cdots ,X_{n1}\right\},\max\left\{X_{12},X_{22},\cdots ,X_{n2}\right\},\;\cdots ,\max\left\{X_{1m},X_{2m},\cdots ,X_{nm}\right\}$$
(5)$$Z^{-}=\left(Z_{1}^{-},Z_{2}^{-},\cdots ,Z_{m}^{-}\right)=(\min\left\{X_{11},X_{21},\cdots ,X_{n1}\right\},\min\left\{X_{12},X_{22},\cdots ,X_{n2}\right\},\cdots ,\min\left\{X_{1m},X_{2m},\cdots ,X_{nm}\right\}$$

The distance from the ith scheme to the positive and negative ideal solutions is shown in Equations (6) and (7), respectively. Then, the relative proximity of each scheme to the positive ideal solution can be calculated, as shown in Equation (8).
(6)$$D_{i}^{+}=\sqrt{\overset{n}{\underset{j=1}{\sum }}{\left(X_{ij}-X^{+}\right)}^{2}}$$
(7)$$D_{i}^{-}=\sqrt{\overset{n}{\underset{j=1}{\sum }}{\left(X_{ij}-X^{-}\right)}^{2}}$$
(8)$$S_{i}^{+}=\frac{D_{i}^{-}}{D_{i}^{-}+D_{i}^{+}},\;0\leq S_{i}^{+}\leq 1$$

When $S_{i}^{+}$ is larger, $D_{i}^{+}$ is smaller and closer to the maximum value. And the scheme can be optimized by sorting $S_{i}^{+}$, as shown in Equation (9).
(9)$$S^{+}=\max\left\{S_{i}^{+}\right\}\;$$

Due to the weighted factor $w_{j}$ reflecting the relative importance of the index, and the reasonable degree of its value affecting the result of the optimal scheme, experts will be too subjective to determine $w_{j}$ with experience, while the entropy weight method can avoid subjective factors. The sample matrix $Z_{ij}^{\prime }$ is normalized to $Z^{\prime }$, as shown in Equation (10). The entropy $E_{j}$ of the normalized value of the jth index is shown in Equation (11). Among them, the $\alpha =1/\ln\left(m\right)$ is constant, m are solutions.
(10)$$Z_{ij}^{\prime }=\frac{Y_{ij}}{\sum_{i=1}^{m}Y_{ij}}\;$$
(11)$$E_{j}=-\alpha \overset{m}{\underset{i=1}{\sum }}Z_{ij}^{\prime }\ln\left(Z_{ij}^{\prime }\right)$$

Equation (12) can calculate the entropy weight of different indicators; if the entropy weight is greater, the significance of the index will be larger.
(12)$$w_{j}=\frac{1-E_{j}}{\sum_{i=1}^{n}\;\left(1-E_{j}\right)}\;$$

## 3. Experimental Design

### 3.1. Materials and Equipment

In order to prepare CCFRPF/PC consumables, CFRPF manufacturing equipment was designed with the melt impregnation process, as shown in Figure 2. The device can make CFRPF by wrapping resin on the surface of continuous raw fiber. The device mainly includes an unwinding module, yarn-spreading module, feeding module, impregnation module, sizing module and winding module. The unwinding module is used to provide continuous raw fiber and tensioning force for the whole preparation process, the yarn-spreading module is used to unfold the continuous fiber, which makes it more conducive to infiltration; the feed module uses a screw extruder (Jiangsu faygo union machinery Co., LTD., Suzhou, China) to provide resin material for the impregnation module; the function of the impregnation module is to cover and infiltrate the continuous raw fiber with high temperature melting resin; the sizing module and winding module ensure the stable and uniform diameter of the formed CFRPF and winding.

The tensile strength of the CFRPF prepared was tested using a universal test machine (XDL-100K, Xinhong Test Machine Factory, Yangzhou, China). CFRPF diameter and roundness were measured by ultra-deep three-dimensional optical microscope (Leica-DM6A). The PC-110 granular material (Taiwan chi mei, Taiwang, China) and the raw carbon fiber T300 (Japan toray) were selected as test materials, and the performance parameters are shown in Table 2 and Table 3.

### 3.2. Performance Indexes and Characterization of Continuous Fiber-Reinforced Prepreg Filament

The CFRPF must meet the requirements of diameter, roundness and minimum curvature before it is used for printing. These performance indexes determine the CFRPF whether it can pass through the printing nozzle smoothly and affect the printing quality of the printed sample. Additionally, if the tensile strength of the monofilament is higher, the enhancement effect on the mechanical strength of the 3D-printed parts will be stronger. Detailed parameter requirements are shown in Table 4.

Diameter and roundness are the main parameters of the shape for CFRPF. In the CFRPF printing process, if the diameter is greater than 0.5 mm, the width of the printed CFRPF is greater than the diameter of the nozzle, which generates excess material that remains in the nozzle and causes blockage. Although the diameter is too small or the roundness error is large, there will be a gap between the CFRPF and the base material or the resin material will overflow, reducing the printing quality and mechanical properties of the printed parts.

N groups were randomly selected from the prepared CFRPF as experimental samples. The diameter is observed and measured by ultra-depth optical microscope, each sample is measured 3 times at a rotation angle of 60°, and the average value is taken as the diameter of the sample, shown in Figure 3a. Then the average value of the data of the N groups obtained by measurement is used as the diameter of CFRCF. Equation (13) is the measurement and calculation method for the diameter of the section.
(13)$$D=\frac{\sum_{1}^{3}L_{i}}{3}\;$$

Calculate the difference value between the maximum diameter and the minimum value on a section and take half of the maximum difference value as the roundness of the section. Additonally, the method of measuring and calculating the roundness of a single section is shown in Figure 2b and Equation (14).
(14)$$\varphi =\frac{d_{\max}-d_{\min}}{2}$$

The minimum curvature radius of CFRPF refers to the minimum bending radius that can be reached without failure or fracture during the CFRPF bending process. In the CFRPF printing process, it needs to be sent to the printing nozzle by the feeding tube, tank chain and other guiding parts. During the transmission process, the CFRPF may break due to bending. Therefore, it is necessary to measure the minimum radius of curvature to reduce fracture failure. Different devices have different requirements on the minimum curvature radius of CFRPF. In this paper, for the self-developed CFRPC printing equipment [24], the minimum curvature radius of the CFRPF is determined to be 20 mm. In order to make CFRPF better meet the requirements of different printing equipment, a sufficient margin is usually reserved for the minimum curvature radius, minimum curvature required radius is $r_{\min}<15\;\mathrm{mm}$.

The test method for the minimum radius of curvature of CFRPF is shown in Figure 4. One end of the CFRCF is fixed and passed between the cylindrical pin and the smooth plane. When bending into a circle with a radius of 20 mm, record the scale value as $L_{1}$ at this time. Then, the CFRPF is slowly pulled along the direction shown in the figure until the crack of CFRPF and the scale $L_{2}$ is recorded. The distance difference D between the two moments is the difference between two circumference lengths, and the minimum curvature radius of CFRPF is shown in Equation (15).
(15)$$r_{\min}=R-\frac{L_{2}-L_{1}}{2\pi }$$

The mechanical properties of CFRC 3DP parts are mainly determined by continuous fiber filaments, so the monofilament tensile strength index of CFRPF is of great significance for evaluating the quality of CFRPF. The tensile strength of monofilaments was measured by the tensile testing machine. The method of calculation of the tensile strength of the monofilament is shown in Equation (16), $T_{\max}$ is the maximum tension, *S* is the cross-sectional area, and *d* is diameter of CFRPF.
(16)$$\sigma =\frac{T_{\max}}{S}=\frac{4T_{\max}}{\pi d^{2}}$$

A CFRCF sample with a length of 160 mm was taken, and two pieces of paper with a certain strength of 30 mm × 30 mm were used at both ends to clamp it, respectively. At the same time, the CFRCF was ensured to be located on the axis of the specimen with an effective length of 100 mm. Then it was fixed with AB epoxy resin adhesive and stood for 24 h before the tensile test. As shown in Figure 5.

### 3.3. Selection of Process Parameters

The key parameters such as impregnation mold temperature, drawing mold temperature and traction speed affect the preparation process of CFRCF. The temperature of the impregnation mold is controlled to allow the melting cavity to melt the resin. The fluidity of the infiltrated resin increases with increasing temperature, but overtemperature then causes degradation of the resin material. According to the single factor test, when the impregnation mold temperature is below 290 °C and the traction speed is increased to 3 m/min, the surface quality of CFRPF is significantly worse. It is also easy to cause the raw fiber to break in the melted resin cavity. Analysis shows that the resin has poor fluidity and high viscosity, so the melting cavity temperature of the impregnation mold is increased. Set its temperature levels to 295 °C, 300 °C and 305 °C.

The outlet mold temperature is used to control the temperature of the sizing module to ensure the surface forming quality of the CFRPF, such as diameter and roundness. If the temperature of the outlet mold is too high, it will lead to excessive melting of the resin on the surface of CFRCF, which is not conducive to styling, and further make the diameter error of CFRCF fluctuate greatly. However, if the temperature is too low, the excess resin material wrapped on the surface of the CFRCF cannot be removed from the exit mold in time, which makes it easy to cause CFRCF fracture. From experience, the temperature of the outlet mold is 10–15 °C lower than that of the impregnation mold, so the temperature levels of the outlet mold are set at 285 °C, 290 °C and 295 °C.

The low winding speed of the composite reduces the efficiency of composite preparation. However, the fusion cavity of the experimental device is small, the high winding speed affects the coating effect of the matrix, resulting in the composite material not being used for printing. In summary, the winding speed is set at 1–2 m/min. The impregnation mold is fixed, and the winding speed mainly affects the impregnation time and the production efficiency of CFRPF. With an increase in the winding speed, the production efficiency is improved, but the continuous fiber filaments stay in the melting cavity of the impregnation mold for a short time, causing the impregnation effect to decrease. Excess winding speed leads to excessive resistance of continuous fiber filaments during impregnation, which may lead to fiber fracture. After the preliminary test, it was found that the continuous fiber filaments began to break slightly in the impregnation mold when the traction speed reached 3 m/min, so the horizontal design of the traction speed was 1.0 m/min, 1.5 m/min and 2.0 m/min.

## 4. Test Design and Scheme Verification

Different level values were selected within a reasonable range of the optimization factor parameters to carry out orthogonal experiments. The table of values of three factors and three levels designed using SPSS 22, the experimental scheme, is shown in Table 5. Additionally, the orthogonal experimental design scheme was developed as shown in Table 6. In the table, A is the temperature of the outlet mold, the unit is °C; B is the temperature of the impregnation mold, the unit is °C; and C is the winding speed, the unit is m/min.

The CCFRPF/PC was prepared according to the experimental scheme, as shown in Figure 6. The influences of the above three factors on the diameter, roundness, minimum curvature radius and tensile strength of monofilament were investigated. In order to more truly express the performance of the materials in each group of tests, 10 groups of measurement data were randomly selected for each performance index, and the average value was taken as the result. The experimental results are shown in Table 7.

The SPSS multivariate analysis of variance was used for the orthogonal test results in Table 7, and the results shown in Table 8. The analysis results showing that Pillai’s trace, Hotelling’s trace and Roy’s maximum root values of Pillai’s trace and Roy’s maximum root values of Pillks’ Lambda (λ) are all large for the detection of the significance of the model effect of the outlet mold temperature, the impregnation mold temperature and the traction speed, and the value of Wilks’ Lambda is equal to zero, which shows that the interaction of the multivariable of the preparation process parameters of the CFRPF contributes a lot to the model. Secondly, the significance of each effect is less than 0.05, indicating that not only the interaction terms of the process parameters have a greater contribution to the model, but also that a single process parameter variable has a significant impact on the model. The preparation parameters of CFRCF/PC are further proven to be reasonable and effective.

## 5. Discussion and Verification of Results

### 5.1. Discussion of the Results

According to the results of the CFRPF performance index in Table 7, the sample matrix Y is established. Equation (10) is used to calculate the normalized matrix $Z^{\prime }$, and the entropy $E=\left[0.926,0.978,0.999,0.998\right]$ and weight $w=\left[0.750,0.221,0.008,0.022\right]$ are calculated from Equations (11) and (12), respectively. The weight value w is substituted into Equations (2) and (3) to calculate the normalized matrix E considering the weighting factors. The distance $D_{i}^{+}$ and $D_{i}^{-}$ of each orthogonal scheme from the positive and negative ideal solutions and the relative proximity $s_{i}^{+}$ are obtained by calculation, and the results are shown in Table 9.

Based on an orthogonal experimental design of nine groups of process parameter combination, it can be seen from the sequence of relative proximity calculation results in entropy weight TOPSIS method that the optimal combination obtained by the weight coefficient determined by the entropy weight method is the 7th group, that is, the outlet mold temperature is 295 °C, the impregnation mold temperature is 300 °C, and the winding speed is 1.5 m/min.

Furthermore, the range analysis on $s_{i}^{+}\;$which is of entropy weight TOPSIS method optimization results of nine groups. On account of calculating numerical approach, the relative proximity degree is $s_{i}^{+}=\left\{19,\;21,\;20,\;25,\;24,\;24,\;27,\;22,\;21\right\}\;$after amplifying $s_{i}^{+}$, and analysis results as shown in Table 10. In the table, $K_{i}$, ${\overset{¯}{K}}_{i}$ and R represent, respectively, the sum, mean and range of relative proximity under trials at each level of various factors.

.

The primary and secondary relationship of each factor was determined on relative proximity by range R, and the primary and secondary level of factors was A>C>B. It can be seen that the outlet mold temperature has the greatest influence on the relative proximity index, followed by the winding speed, and the impregnation mold temperature has the least influence. The optimal combination of all factors is A1C1B3.

The influence of various factors on relative proximity is shown in Figure 7, it can be seen that the optimal level of outlet mold temperature and winding speed is the first level, and the impregnation mold temperature is the third level. Therefore, the combination of optimum index process parameters of the CCFRPF/PC should be as follows: the outlet mold temperature should be 285 °C, the impregnation mold temperature should be 305 °C, and the winding speed should be 1 m/min.

### 5.2. Experimental Verification

CCFRCF/PC was prepared using the combination of optimized process parameters. The prepared consumables were measured for each performance index parameter, and each parameter was measured 10 times, take average value as the final performance index, and the results are shown in Table 11. According to the measurement results, it can be seen that the diameter can be controlled at about 0.375 mm, which is at an excellent level compared with the diameter in the nine groups of orthogonal test schemes. And the roundness and minimum curvature radius are moderately above the horizontal. Therefore, in this paper, the optimization strategy of the process parameters is feasible. The optimum preparation parameters that meet the performance index requirements are as follows: the outlet mold temperature is 285 °C, the impregnation mold temperature is 305 °C, and the winding speed is 1 m/min. In this case, the diameter, roundness, minimum curvature radius and tensile strength of 0.375 mm, 29.4 μm, 9.775 mm and 1298 MPa were achieved, respectively.

## 6. Conclusions

This paper mainly studies the CFRCF/PC preparation process, and a preparation device was designed and manufactured by resin melt impregnation theory. The performance indices and the process parameters of CFRCF were analyzed to meet the printing requirements of CFRC 3DP. The orthogonal test and the TOPSIS method of entropy weight were used to optimize the preparation process parameters of CFRPF/PC. The results show that in the preparation of CCFRPF/PC, the diameter performance index weight is the largest, about 0.75 mm. The combination of optimal process parameters of mold exit temperature, impregnated mold temperature and winding speed was 285 °C, 305 °C and 1 m/min, respectively. In this case, the diameter, roundness, minimum curvature radius and tensile strength of 0.375 mm, 29.4 μm, 9.775 mm and 1298 MPa were achieved, respectively. In addition, the process optimization strategy studied in this paper can provide methods for the preparation of more types of CFRPF.

## Figures and Tables

**Figure 1 polymers-15-00607-f001:**
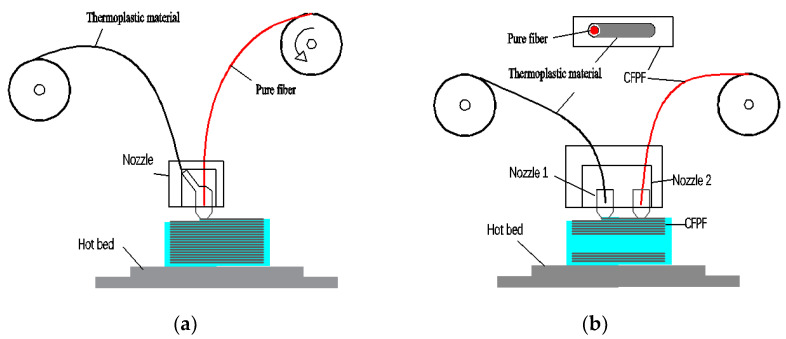
Schematic diagram of CFRC 3DP. (**a**) Type of “co-extrusion”. (**b**) Type of “independent extrusion”.

**Figure 2 polymers-15-00607-f002:**
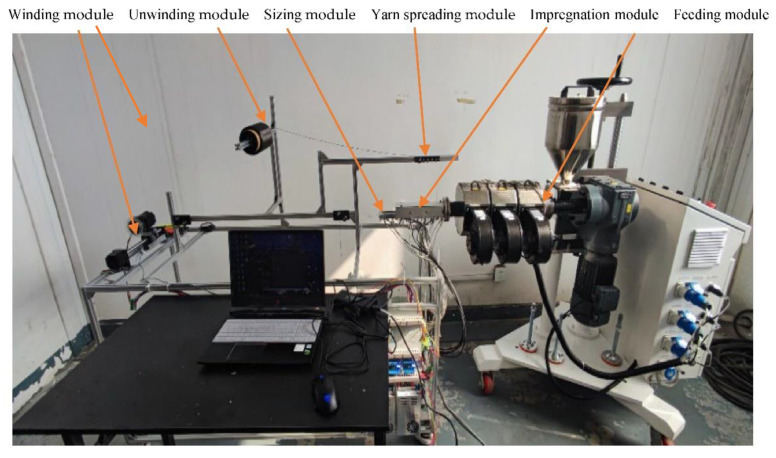
Equipment to prepare CFRPF.

**Figure 3 polymers-15-00607-f003:**
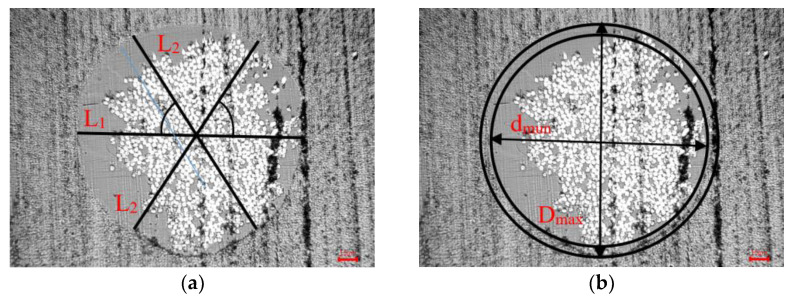
Measuring method for the diameter and roundness of CFRPF. (**a**) Measurement method of diameter. (**b**) Measurement method of roundness.

**Figure 4 polymers-15-00607-f004:**
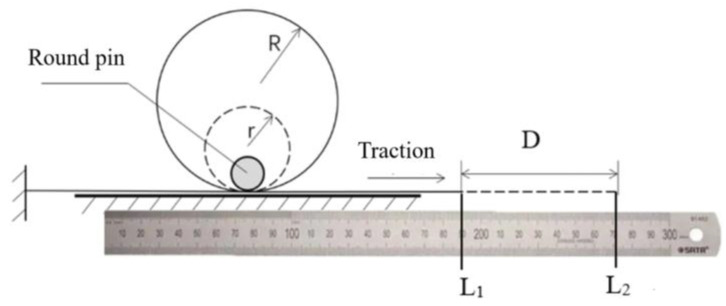
Measuring method of the minimum curvature radius.

**Figure 5 polymers-15-00607-f005:**
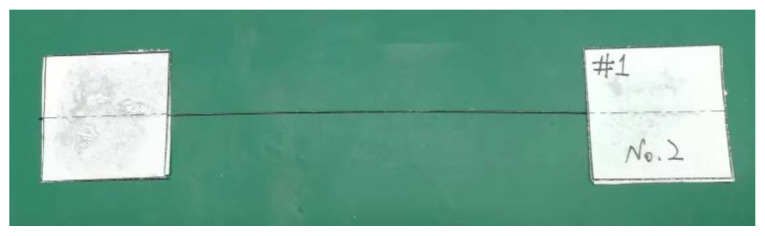
Preparation of CFRPF monofilament tensile samples.

**Figure 6 polymers-15-00607-f006:**
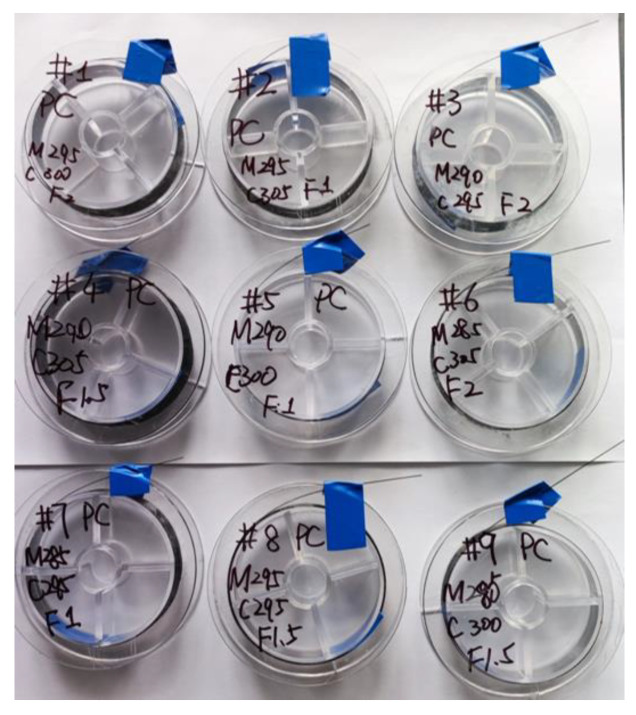
CFRPF prepared by orthogonal test.

**Figure 7 polymers-15-00607-f007:**
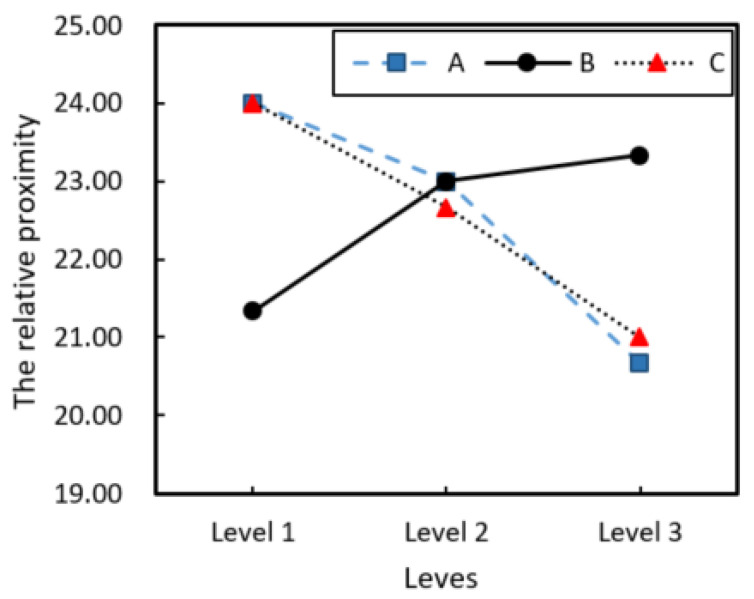
The influence of various factors on relative proximity.

**Table 1 polymers-15-00607-t001:** Material parameters of the CFRPF at MarkForged company.

Reinforcing Material	Diameter (mm)	Density (g/cm^3^)	Tensile Modulus (GPa)	Flexural Modulus (GPa)
Continuous carbon fiber	0.4	1.2	60.0	51.0
Continuous glass fiber	0.3	1.5	21.0	22.0
Continuous Kevlar fiber	0.3	1.2	27.0	26.0

**Table 2 polymers-15-00607-t002:** Performance parameters of PC-110 granular material.

Performance	Attribute Value	Test Criteria
Melting point (°C)	240	-
Yield tensile strength (MPa)	61.8	ASTM D882
Tensile strength at break (MPa)	75	ASTM D882
Density (g/cm^3^)	1.20	ASTM D1505

**Table 3 polymers-15-00607-t003:** Carbon fiber performance parameters (T300).

Performance	Attribute Value	Test Criteria
Tensile strength (MPa)	3530	TY-030B-01
Tensile modulus (GPa)	230	TY-030B-01
Strain (%)	1.5	TY-030B-01
Density (g/cm^3^)	1.76	TY-030B-02
Diameter of single fiber (μm)	7	ASTM D1505

**Table 4 polymers-15-00607-t004:** Parameter requirements of CFRPF for printing.

Performance	Technical Parameters
Diameter (mm)	0.30–0.50
Roundness (mm)	<0.03
Minimum curvature radius (mm)	<15.00

**Table 5 polymers-15-00607-t005:** Factor level value table.

Influence Factor	Level 1	Level 2	Level 3
Impregnation mold temperature (°C)	295	300	305
Outlet mold temperature (°C)	285	290	295
Winding speed (m/min)	1.0	1.5	2.0

**Table 6 polymers-15-00607-t006:** Orthogonal test scheme.

Serial Number	Influencing Factors and Levels			
	A	B	C	Null Columns
1	295	300	2	1
2	295	305	1	2
3	290	295	2	2
4	290	305	1.5	1
5	290	300	1	3
6	285	305	2	3
7	285	295	1	1
8	295	295	1.5	3
9	295	300	1.5	2

**Table 7 polymers-15-00607-t007:** Summary table of orthogonal test results.

Serial Number		Performance Index of CCFRPF/PC		
	Diameter (mm)	Roundness (μm)	Minimum Radius of Curvature (mm)	Tensile Strength of Monofilament (MPa)
1	0.348	45	10.562	1441.723
2	0.352	40	10.387	1277.539
3	0.343	35	10.180	1368.821
4	0.377	25	10.036	1149.243
5	0.358	20	9.113	1159.534
6	0.366	57	11.288	1161.089
7	0.370	34	10.084	1028.869
8	0.370	24	9.703	1269.590
9	0.375	25	9.559	1329.583

**Table 8 polymers-15-00607-t008:** Results of the multivariate variance analysis of CFRPF performance indicators.

Multiple Variable Test ^a^						
Effect		Value	F	Hypothetical df	Error df	Significance
Intercept	Pillai’s trace	1.000	195,055.074 ^b^	2.000	1.000	0.002
	Wilks’ Lambda (λ)	0.000	195,055.074 ^b^	2.000	1.000	0.002
	Hotelling’s trace	390,110.148	195,055.074 ^b^	2.000	1.000	0.002
	Roy’s maximum root value	390,110.148	195,055.074 ^b^	2.000	1.000	0.002
Outlet mold temperature	Pillai’s trace	1.983	115.487	4.000	4.000	0.000
	Wilks’ Lambda (λ)	0.000	67.585 ^b^	4.000	2.000	0.015
	Hotelling’s trace	316.359	0.000	4.000	0.000	0.000
	Roy’s maximum root value	240.616	240.616 ^c^	2.000	2.000	0.004
Impregnation mold temperature	Pillai’s trace	1.939	31.685	4.000	4.000	0.003
	Wilks’ Lambda (λ)	0.000	22.555 ^b^	4.000	2.000	0.043
	Hotelling’s trace	128.099	0.000	4.000	0.000	0.000
	Roy’s maximum root value	109.934	109.934 ^c^	2.000	2.000	0.009
Winding speed	Pillai’s trace	1.981	102.594	4.000	4.000	0.000
	Wilks’ Lambda (λ)	0.000	93.571 ^b^	4.000	2.000	0.011
	Hotelling’s trace	681.392	0.000	4.000	0.000	0.000
	Roy’s maximum root value	625.930	625.930 ^c^	2.000	2.000	0.002

^a^ Design: intercept + outlet mold temperature + impregnation mold temperature + winding speed. ^b^ Exact statistics. ^c^ Statistics are the upper limit of F, which produces the lower limit at the level of significance.

**Table 9 polymers-15-00607-t009:** Relative proximity under the weight coefficient scheme of the entropy weight method.

Serial Number	Normalized Diameter	Normalized Roundness	Normalized Minimum Curvature	Normalized Tensile Strength	$D_{i}^{+}$	$D_{i}^{-}$	$s_{i}^{+}\;(10^{-3})$	Sequence
1	0.107	0.216	0.317	0.385	214.278	214.294	500.019	9
2	0.160	0.243	0.323	0.341	189.869	189.885	500.021	7
3	0.040	0.278	0.329	0.365	203.440	203.456	500.020	8
4	0.494	0.390	0.334	0.307	170.796	170.813	500.025	2
5	0.240	0.487	0.368	0.310	172.326	172.342	500.024	4
6	0.347	0.171	0.297	0.310	172.557	172.574	500.024	3
7	0.400	0.286	0.332	0.275	152.901	152.917	500.027	1
8	0.400	0.406	0.345	0.339	188.687	188.704	500.022	5
9	0.467	0.390	0.350	0.355	197.606	197.623	500.021	6

**Table 10 polymers-15-00607-t010:** The results of range analysis on relative proximity degree $s_{i}^{+}$.

Index	A	B	C
$K_{1}$	72.00	64.00	72.00
$K_{2}$	69.00	69.00	68.00
$K_{3}$	62.00	70.00	63.00
${\overset{¯}{K}}_{1}$	24.00	21.33	24.00
${\overset{¯}{K}}_{2}$	23.00	23.00	22.67
${\overset{¯}{K}}_{3}$	20.67	23.33	21.00
R	3.33	2.00	3.00
Factor primary and secondary level	A > C > B		
Optimal level	A1	B3	C1
Optimal combination	A1C1B3		

**Table 11 polymers-15-00607-t011:** The CCFRCF/PC performance index in optimized preparation process parameters.

Serials	Diameter (mm)	Roundness (μm)	Minimum Radius of Curvature (mm)	Tensile strength (MPa)
1	0.372	27.000	9.452	1391.373
2	0.387	31.000	9.934	1278.735
3	0.365	29.000	9.511	1264.629
4	0.379	29.000	10.664	1297.167
5	0.385	35.000	9.865	1362.837
6	0.367	28.000	10.189	1350.434
7	0.371	31.000	9.268	1106.071
8	0.378	26.000	9.713	1261.176
9	0.375	25.000	9.456	1295.723
10	0.375	33.000	9.698	1368.210
Mean	0.375	29.400	9.775	1297.636
Standard deviation	0.007	2.970	0.390	77.640

## Data Availability

The data are contained within the article.

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
