# Peer review of "Preparation and Process Parameter Optimization of Continuous Carbon Fiber-Reinforced Polycarbonate Prepreg Filament"

_polymers, 2023, doi:10.3390/polym15030607_

Round 1

Reviewer 1 Report

The paper examines quality of filamanets through a statistical optimisation of process parameters. The subject of the paper presents a definite interest to research community. Unfortunately, in its current state the text is barely readable - it is littered with incorrect grammar and awckward phrasing making it very difficult to read and comprehend the message. Once corrected, there may be questions on the choice of relevant parameters, underlaying assumptions for the desing of experiments, assessment of strength, independent strain measurements in the tests, etc. However, this can only be addressed once the text is in acceptable form.

Author Response

Response:

I'm sorry that my expression confused you. The whole article has been revised and polished, and all revisions to the manuscript have been marked up using the “Track Changes” function.

Thanks for your advice. In this paper, the continuous fiber reinforced prepreg filament was only considered to be able to achieve continuous fiber reinforced composite 3D printing on the choice of relevant parameters. But the relevant research will be carried out in the future for your suggestion. Thank you again.

Reviewer 2 Report

Dear authors and editor

My comments:

1. I think there is no necessary to mention the names of universities in the introduction and literature review (Beijind and Jiatong).

2. If possible, mention some examples of specific applications of the CCFPF/PC composite material.

3. What is the pressure range achieved in the implementation mold.

4. The numbers of the used references is few.

5. If the SEM image from the fracture section of the impregnated fiber or tensile test sample is available, added to the article (in order to the observation the voids and interface of fiber and polymer)

Best Regards

Round 2

Reviewer 1 Report

The paper still needs extensive editing of grammar and awkward expressions - it is not suitable for the publication in the current state. On the technical side, the paper misses the full description of the tests, such as the discussion of independent strain measurements in tensile tests. On the conceptual side, the selection of the performance indexes requires further justification. It could be argued that porosity and fibre volume fraction of filament is far more important than roundness of the filament - the filaments will anyway be reshaped under the nozzle of the printer. 

Round 3

Reviewer 1 Report

The new edition addresses most of the questions, remarks suggested previously.